# Structure Determination of Felodipine Photoproducts in UV-Irradiated Medicines Using ESI-LC/MS/MS

**DOI:** 10.3390/pharmaceutics15020697

**Published:** 2023-02-19

**Authors:** Kohei Kawabata, Miya Kohashi, Shiori Akimoto, Hiroyuki Nishi

**Affiliations:** 1Faculty of Pharmacy, Yasuda Women’s University, Yasuhigashi 6-13-1, Asaminami-ku, Hiroshima 731-0153, Japan; 2Akimoto Pharmacy, Akimoto Pharmacy Corporation, 7-17, Akama-cho, Shimonoseki 750-0007, Japan

**Keywords:** felodipine, photodegradation, photodimerization, photooxidation, photoproduct, HPLC, ESI-LC/MS/MS

## Abstract

Dihydropyridine drugs are well known as photodegradable pharmaceuticals. Herein, we evaluate the photostability of felodipine (FL) medicine (Splendil^®^ (SPL) tablets) and its altered forms (powders and suspensions). FL is a type of dihydropyridine drug, but its photochemical behavior is unknown. FL contents after ultraviolet light (UV) irradiation for 24 h were monitored using high-performance liquid chromatography (HPLC). Values of the residual amounts of FL in UV-irradiated SPL powders and suspensions were 32.76 ± 4.88% and 0.79 ± 0.74%, respectively, with the generation of two photoproducts (FL photoproduct 1 and 2). To identify the chemical structures of these photoproducts, electrospray ionization liquid chromatography mass spectrometry (ESI-LC/MS/MS) analysis was performed. Based on their mass-to-charge ratio values and fragment patterns, it was proposed that FL photoproduct 1 was a pyridine derivative and FL photoproduct 2 was an FL dimer. Interestingly, generation rates of FL photoproduct 1 and 2 were dependent on the presence of the aqueous media. The photodimerization of FL was induced in UV-irradiated SPL suspensions. This is the first report evaluating the photostability of SPL tablets and its altered forms and estimating FL photoproducts induced by UV irradiation in the formulation of SPL.

## 1. Introduction

Photoexposure induces a change of both the quality and quantity of photosensitive pharmaceuticals by photodegradation. Yamashita et al. showed the color change of several medicines without a press through package (PTP) sheet under LED lighting and fluorescent lighting [1]. Furthermore, active pharmaceutical ingredients (APIs) of several normal plain tablets, such as amlodipine, nifedipine and nilvadipine, were photodegraded when these tablets were pulverized and photoirradiated [2]. In a clinical situation, there are several cases where tablets are crushed and packaged followed by storage in the houses of patients. Our previous reports also showed that long-term storage (for up to 90 days) of crushed amlodipine oral disintegration tablets in the package near the window of a house, where it is exposed to sunlight irradiation, resulted in a decrease in their APIs [3]. The decrease in APIs means the weakening of the beneficial effect of a medicine. So, the photostability evaluation of a photosensitive medicine and its altered forms (powders and suspensions) is an important issue for their safe use.

Photodegradation of pharmaceuticals results in the generation of their photoproducts. In the case of dihydropyridine drugs, the dihydropyridine ring is easily oxidized, followed by the generation of a pyridine derivative [4,5,6,7]. Dihydropyridine drugs have been used for the reduction of high blood pressure but are well known as photosensitive drugs. Nifedipine is one of such dihydropyridine drugs and its main photoproduct is known as a pyridine derivative with a nitroso group [8,9,10]. This photoproduct was generated by the elimination of a hydrogen atom in the dihydropyridine ring by one oxygen atom of the nitro group followed by the additional elimination of a hydrogen atom. Interestingly, its ability to inhibit calcium channels is quite weak or non-existent compared with that of nifedipine [11,12,13]. On the other hand, this photoproduct showed the antioxidative potency [14]. These studies indicate that different natures of photosensitive pharmaceuticals lead to different photoproducts. On the contrary, photoexposure might lead to the expression of an adverse effect. For example, dacarbazine is photoconverted to 5-diazoimidazole-4-carboxamide which induces vascular pain even though dacarbazine has no effect [15]. Several reports including our previous studies showed that the toxicological potencies of photosensitive pharmaceuticals were induced by ultraviolet light (UV) irradiation through the generation of their photoproducts [16,17,18,19,20,21].

Stability test guidelines have been established by the International Conference on Harmonization of Technical Requirements for Regulation of Pharmaceuticals for Human Use (ICH) (ICH-Q1A, Stability testing of new drug substances and products) for the evaluation of the stability of pharmaceuticals [22]. The ICH-Q1A gives the test conditions, such as the long-term storage testing (under the condition of 25 ± 2 °C/60 ± 5% relative humidity (RH) or 30 ± 2 °C/65 ± 5% RH for 12 months), the intermediate storage testing (under the condition of 30 ± 2 °C/65 ± 5% RH for 6 months) and the accelerated storage testing (under the condition of 40 ± 2 °C/75 ± 5% RH for 6 months). Furthermore, for the determination of intrinsic stability, stress tests are carried out under severe conditions (heat, humidity and photoexposure). In the case of photostability evaluation, test guidelines (ICH-Q1B, Photostability testing of new drug substances and products) have also been established [23]. Obtained results are described in the interview form and used as the evidence for the determination of the type of a final package and a dosage form of test pharmaceuticals. However, photostability tests focused on medicines which are out of a PTP sheet or changed to powders and suspensions are not performed. For most tablets, the effect of changes to dosage form on their photostability is unclear. With this in mind, in this study, we first evaluate the effects of crush and suspension on the photostability of API. We believe this information is important for their safe use.

In the present study, we performed the photostability evaluation of Splendil^®^ 2.5 mg (SPL) tablets whose API is felodipine (FL, Figure 1). SPL tablets is an original medicine of FL tablets. FL is a type of dihydropyridine drug and chemically ethyl methyl (4RS)-4-(2,3-dichlorophenyl)-2,6-dimethyl-1,4-dihydropyridine-3,5-dicarboxylate. Its molecular formula and molecular weight are C_18_H_19_Cl_2_NO_4_ and 384.25 g/mol. FL is widely used and can simultaneously reduce the blood pressure of patients and significantly increase the plasma auxin concentrations [24]. This compound is a crystalline powder with very low water solubility [25], resulting in poor oral bioavailability [26]. Tres et al. reported the detection of FL in the environment using fluorescence techniques [27]. This is the first report focused on the photochemical behavior of FL tablets. Firstly, residual amounts of API in UV-irradiated SPL tablets and its altered forms (powders and suspensions) were evaluated by high-performance liquid chromatography (HPLC). Next, chemical structures of two photoproducts were elucidated by electrospray ionization tandem mass spectrometry (ESI-LC/MS/MS) analysis. Finally, the photodegradation mechanism of SPL tablets was discussed based on the obtained results. As a result of this study, it was clarified that the photostability of SPL tablets decreased by the change of the dosage form, and two FL photoproducts, which were a pyridine derivative and an FL dimer, were identified in the UV-irradiated SPL powders and suspensions. Interestingly, generation rates of these photoproducts were affected by the dosage forms of SPL tablets, suggesting that the form of medicines might contribute to the photodegradation mechanism of APIs.

## 2. Materials and Methods

### 2.1. Materials

SPL tablets (Splendil^®^ 2.5 mg tablet, AstraZeneca plc, Cambridge, UK) were purchased from a commercial source. Formic acid (99%) and methanol (MeOH, 99%) were obtained from Fujifilm Wako Pure Chemical Corporation (Osaka, Japan). FL drug substance used as a standard substance was obtained from Tokyo Chemical Industry Co., Ltd. (Tokyo, Japan). All reagents and organic solvents were of special or HPLC grade. Milli-Q (18.2 Ω/cm) water was prepared by using a Milli-Q water purification system (Merck, Darmstadt, Germany).

### 2.2. Sample Preparation

One SPL tablet was taken out from the PTP sheet and utilized as a tablet sample. SPL tablets were also crushed or suspended to prepare test samples (a powder sample and a suspension sample). For the preparation of a suspension sample, one SPL tablet was suspended in 50 mL of Milli-Q water and sonicated for 10 min. These samples were exposed to a light source for 24 h. Control samples were prepared with the same procedures but covered with aluminum foil to protect them from photoexposure. UV-irradiated tablet samples and powder samples were dissolved in 100 mL of 50% (*v*/*v*) MeOH and sonicated for 5 min for extraction. These solutions were filtered through a membrane filter (0.45 μm, Merck), and the filtrate was analyzed by HPLC. UV-irradiated suspension samples were added to MeOH to make up 100 mL exactly. The residue was removed the same as in tablet samples and powder samples, and the filtrate was subjected to HPLC analysis. In addition, for the purpose of the evaluation of HPLC method validation, FL drug substance was dissolved in 50% (*v*/*v*) MeOH and utilized as a standard solution. All experiments were carried out in quadruplicate.

### 2.3. Irradiation Experiment

UV irradiation experiments were performed in a light cabinet equipped with a black light lamp (FL20S BLB, Toshiba, Tokyo, Japan). This lamp emits long-wavelength UV and very little visible light and the most abundant wavelength is 365 nm, which is a component of sunlight. UV irradiation intensity was 300 µW/cm^2^/sec measured by a digital radiometer with a 365 nm sensor (UVX-36, UVP, Upland, CA, USA). UV irradiation experiments were carried out at temperature: 20 °C, distance from a light source: about 20 cm (tablets and powders) and 15 cm (suspensions), water depth: 3 cm (suspensions). Irradiation times for SPL tablets and its altered forms were up to 24 h.

### 2.4. HPLC Analysis

HPLC analysis was performed with a Prominence series composed of an LC-20AB, a SIL-20AC autosampler, an SPD-M20A photodiode array (PDA) detector with a LCsolution software, a CBM-20A system controller, a DGU-20A3 degasser and a CTO-20A column oven (Shimadzu Corp., Kyoto, Japan). An analytical column (Shim-pack Arata C18 column, 4.6 × 150 mm, particle size 5 µm, Shimadzu Corp.) was kept at 40 °C. Isocratic separations were performed using a mobile phase which is a mixture of MeOH and 0.1% (*v*/*v*) formic acid (8:2, *v*/*v*). This solvent was appropriate as a mobile phase for this experiment because the analytical time was within 20 min and the peak separation was good. The flow rate was maintained at 0.5 mL/min, and the injection volume was 10 µL. The detection wavelength was set at 254 nm. Amounts of FL and its photoproducts were shown as the residual rate (%) for amounts before UV irradiation, which were calculated using their peak areas evaluated by HPLC. The UV absorption spectra of FL and its photoproducts were recorded with a PDA detector. UV-irradiated SPL suspensions were injected to the HPLC system, and UV spectra of the peaks of FL and its photoproducts were recorded under the HPLC conditions as mentioned above. In addition, a PDA detector was utilized for the purity check of detected peaks showing that each peak was not overlapped under the above LC condition. 

### 2.5. HPLC Method Validation

The HPLC method of FL in tablets was validated with regard to specificity, the calibration curve, linearity (range), accuracy and precision according to ICH Q2 guideline (Validation of analytical procedure) [28]. The specificity was evaluated by changing HPLC conditions and using purity check function of PDA detector. Calibration curve was acquired by plotting of each concentration (corresponding to 20%, 40%, 60%, 80%, 100% and 120% of the assay solution) versus those peak areas. Limit of quantification (LOQ) was evaluated from the calibration curve (slope and the standard deviation (SD) of the response (σ), LOQ = 10σ/slope). Accuracy was evaluated by recovery testing. Two solutions were prepared as follows: One is a sample solution of SPL tablets, which was dissolved in 50% (*v*/*v*) MeOH and filtrated to remove the residue mentioned above, and then added to a standard solution of FL (FL concentration is known). The other is a sample solution of SPL tablets. After HPLC determination of these two solutions, recovery was calculated by deduction. Precision was evaluated by analyzing replicate tablet samples (*n* = 5). Precision was expressed as the relative standard deviation (RSD) of five replicated assay values.

### 2.6. Structure Identification of FL Photoproducts

Structural identification of FL photoproducts was carried out by a LC-MS system composed of a LC-20AD pump, a SIL-20AC autosampler, a CBM-20A system controller, a DGU-20A5R degasser, a CTO-20A column oven, an FCV-20AH2 valve unit and an LCMS-8040 MS system equipped with ESI as the ionization source (Shimadzu Corp.). The whale system was controlled by a LabSolution software (Shimadzu Corp.). An analytical column (Shim-pack Arata C18 column, 2.0 × 100 mm, particle size 2.2 µm, Shimadzu Corp.) was kept at 40 °C. Isocratic separations were performed using a mobile phase which is a mixture of MeOH and 0.1% (*v*/*v*) formic acid (8:2, *v*/*v*). The flow rate was maintained at 0.1 mL/min, and the injection volume was 10 µL. The mass spectrometer was run in the positive ion mode in the range of 50–1000 mass-to-charge ratio (m/z). The following parameters were applied during the analysis: nebulizer gas flow, 3.0 L/min; drying gas flow, 15.0 L/min; collision-induced dissociation (CID) gas, 230 kPa; desolvation line (DL) temperature, 200 °C; heat block temperature, 200 °C; and collision energy, among −10 and −40 V. Sample solutions prepared from UV-irradiated SPL suspensions were subjected to LC-MS and LC-MS/MS analysis by the scan mode and the results were compared with that of the control sample. Detected peaks only from UV-irradiated samples were analyzed by the product ion scan mode to determine fragment patterns.

### 2.7. Statistical Analysis

Data are expressed as the mean ± SD. Statistical significance between two groups was estimated by the Student’s *t*-test. The threshold for assessing significance were *p* < 0.001 (***) versus the control samples and *p* < 0.001 (^###^) versus the powder samples.

## 3. Results

### 3.1. Photostability of SPL Tablets and Its Altered Forms

In this study, we aim to clarify the photochemical behavior of SPL tablets and its altered forms. In the first step, the evaluation of the validity of the HPLC method was performed. Results are summarized in Table 1. Obtained validation parameters were suitable for the evaluation of FL content.

UV irradiation experiments for SPL tablets and its altered forms were carried out. UV irradiation was performed using a black light. It mainly emits UV-A (most abundant wavelength is 360 nm). UV-A is the main component of UV-range light in sunlight. Furthermore, our previous study showed that the photodegradation induced by UV-A irradiation was comparable to that induced by sunlight irradiation [16]. We think that this study demonstrates the photostability evaluation in the actual conditions as much as possible. Effects of UV irradiation for 24 h on APIs in tablet samples, powder samples and suspension samples of SPL tablets were evaluated by HPLC. Typical chromatograms are shown in Figure 2. In the case of tablet samples, one peak derived from FL was detected at a retention time of 7.9 min. UV irradiation had no effect on SPL tablets because the peak area of FL was not changed, and other peaks were not detected. SPL tablets are film-coated tablets, so their APIs were protected from UV irradiation and the photodegradation was not induced. On the other hand, changing their dosage forms to powders and suspensions resulted in the decrease in photostability as shown in Figure 2D,F. Both the decrease in API and the generation of two photoproducts (FL photoproduct 1 and 2, retention times were 6.0 min and ca. 15 min, respectively) were observed after UV irradiation. This result suggests that FL photoproduct 1 has higher polarity than FL judged from the faster retention time in C18 column. On the contrary, FL photoproduct 2 seems to have lower polarity. Among two altered forms, the photodegradation of APIs in SPL suspensions were more significant because the peak of API was slightly detected. Other photoproducts were not detected in this experiment. In our experiments, the control samples of SPL powders and suspensions did not change, indicating that the degradation of SPL resulted from only UV irradiation, and other factors, such as temperature and humidity, had no contribution on it.

UV spectra of FL and two FL photoproducts recorded by a PDA detector are shown in Figure 3. FL showed a characteristic maximum absorption at 361 nm which was derived from a dihydropyridine moiety. In contrast to FL, both FL photoproduct 1 and 2 showed no absorption between 300 and 400 nm, strongly suggesting that these photoproducts might not have the dihydropyridine moiety. Additionally, the change of the color of SPL tablets, powders and suspensions were not observed by UV irradiation.

The residual amounts (%) of APIs and the generation rates (%) of two FL photoproducts in tablet samples, powder samples and suspension samples are summarized in Figure 4. Residual amounts of APIs in UV-irradiated powders and suspensions (32.76 ± 4.88% and 0.79 ± 0.74%, respectively) were significantly less than those of the control samples (94.69 ± 2.52% and 100.22 ± 0.83%, respectively). Similarly to the HPLC chromatogram, residual amounts of APIs were not influenced by UV irradiation when tablet samples were irradiated. These results indicated that changing the dosage form of SPL tablets to powders and suspensions decreases the photostability because of the loss of a film coating and the enhancement of a light emission efficiency to their APIs, as shown in our previous reports [29]. It is possible that the crushing of SPL tablets for the improvement of medication compliance in a clinical situation might result in the decrease in its beneficial effect when SPL powders are packaged and stored in a place where they are exposed to sunlight.

In addition, the values of the generation rates of FL photoproduct 1 and 2 in UV-irradiated powder samples were 5.63 ± 0.30% and 2.52 ± 0.08%, respectively (Figure 4B). These photoproducts were not detected in UV-irradiated tablet samples. The generation rates of FL photoproducts, which were calculated using their peak areas and those of FL, were less than 10%, probably because of their poor UV absorption at 254 nm as shown in Figure 3. The generation of FL photoproduct 2 was accelerated by the suspension of SPL tablets followed by UV irradiation judged from its higher generation rate (9.91 ± 0.51%). Interestingly, the generation rate of FL photoproduct 1 in the suspension samples (2.32 ± 0.47%) was significantly lower compared with those in the powder samples (5.63 ± 0.30%), while residual amounts of APIs in suspension samples were dramatically low. These results suggest that the generation of FL photoproduct 1 occurs prior to that of FL photoproduct 2 in the absence of an aqueous medium. The FL photochemical reaction induced by UV irradiation to form FL photoproduct 1 and 2 seems to be affected by the aqueous media.

### 3.2. Structural Determination of Two FL Photoproducts

Structural determination of two FL photoproducts detected in HPLC analysis was conducted by subjecting sample solutions prepared from SPL suspensions with or without UV irradiation to the ESI-LC/MS/MS system. The MS spectra, MS/MS spectra and speculated fragmentation of FL, and FL photoproduct 1 and 2 are shown in Figure 5, Figure 6 and Figure 7.

First, analysis of FL was carried out to obtain some information on the structural elucidation of its photoproducts. The *m*/*z* value of a protonated FL was obtained at 384.25 and 386.10, which were isotopic ion peaks derived from two chlorine atoms in the positive ion mode (Figure 5A). Two fragment ions with *m*/*z* values of 352.00 and 338.00, respectively, were detected in MS/MS spectra (Figure 5B). These fragment ions seem to be formed by the loss of a methoxy group or an ethoxy group (Figure 5C).

The *m*/*z* value of protonated FL photoproduct 1 was 382.20 and 384.20 as shown in Figure 6A. FL photoproduct 1 has also two chlorine atoms due to the detection of isotopic ion peaks. The *m*/*z* value of FL photoproduct 1 was smaller than that of FL by 2, which corresponds to the molecular weight of a hydrogen molecule. It is suggested that FL photoproduct 1 is a pyridine derivative of FL, which is chemically 3-ethyl 5-methyl 4-(2,3-dichlorophenyl)-2,6-dimethylpyridine-3,5-dicarboxylate. In the MS/MS spectrum, one fragment ion (*m*/*z* 354.00) was detected (Figure 6B) and seemed to be generated by the elimination of an ethyl group (Figure 6C). It is indicated that the oxidation of a dihydropyridine ring resulted in the decrease in its polarity because FL photoproduct 1 eluted faster than FL as shown in Figure 2, the same as our previous report [3]. In addition, a pyridine derivative showed no characteristic absorption among 300 to 400 nm, which differs from FL as shown in Figure 3B. 

As for FL photoproduct 2, the molecular ion peak of a protonated FL photoproduct 2 (*m*/*z* 769.35) was detected in the MS chromatogram (Figure 7A). Several isotopic ion peaks were detected, and their detection pattern was different from that of FL and FL photoproduct 1. Its MS/MS chromatogram (Figure 7B) showed that FL and FL photoproduct 1 were detected as fragment ions of FL photoproduct 2, whose *m*/*z* values were 384.05 and 382.05, respectively, and their additional fragment ions were generated by the elimination of a methoxy group and an ethoxy group. It is postulated that the structure of FL photoproduct 2 is an FL dimer which is chemically 3,7-diethyl 4a,8a-dimethyl 4,8-bis(2,3-dichlorophenyl)-2,4b,6,8b-tetramethyl-1,5,8,8b-tetrahydrocyclobuta [1,2-b:3,4-b′]dipyridine-3,4a,7,8a(4H,4bH)-tetracarboxylate (Figure 7C). FL dimer has four chlorine atoms, so several isotopic peaks are observed in the MS chromatogram as shown in Figure 7A. Several minor fragment ions (*m*/*z* 737.15, 723.05 and 691.05, respectively) were also designated as product ions generated by the loss of a methoxy group, ethoxy group and both groups. UV spectral analysis clarified that FL photoproduct 2 showed no characteristic absorption among 300 to 400 nm, which is different to FL (Figure 3C), indicating that a photodimerization induced the loss of a dihydropyridine moiety the same as FL photoproduct 1.

### 3.3. Photodegradation Mechanism of FL

As a result of ESI- LC/MS/MS analysis, two FL photoproducts were clarified as a pyridine derivative and a dimer of FL (Figure 1). We thought two FL photodegradation pathways were independent because two FL photoproducts were not generated step by step as shown in Figure 2. This is the first study focused on the photochemical behavior of an FL medicine. It has been reported that oxidation of the dihydropyridine ring was the main photodegradation pathway of various dihydropyridine drugs such as amlodipine and nifedipine in the solid-state [2]. Obtained results in this experiment also indicated that a pyridine derivative of FL was the main photoproduct when SPL powders were UV-irradiated (Figure 2D). Elimination of proton radical is a trigger of the oxidation of a dihydropyridine ring of FL, followed by the elimination of proton from nitrogen atom resulting in the formation of an aromatic moiety. However, in the case of SPL suspensions, the main photoproduct was FL photoproduct 2 due to its higher generation rate compared with FL photoproduct 1 (Figure 4B). It is proposed that two FL molecules might be bound to form FL photoproduct 2 through the bridging of each dihydropyridine ring by a π-π interaction. These results make it possible to estimate that the photodimerization of FL might be easily induced in aqueous media, compared with in the powder form, because FL molecules could move freely and there is less distance between FL molecules. A promotion of the photodimerization resulted in the suppression of FL oxidation and the decrease in the generation rate of FL photoproduct 1. To the best of our knowledge, this is the first study showing that the dosage form contributes to the formation of photoproducts in addition to the photodegradation rates. It is possible that other factors except for an aqueous media, such as irradiation efficiency and oxygen molecules, might have a crucial role in the photodegradation pathway. In addition, FL photodegradation might be affected by the pH value, which is derived from the proton ion content and dominates the state of ionization of FL in the suspension form, so further research is needed to evaluate the FL photodegradation under several conditions.

Moreover, there are several additives in SPL tablets. Titanium dioxide (TiO_2_) and iron (Ⅲ) oxide (Fe_2_O_3_) are utilized as coating agents, but they are well-known photocatalysts. They are excited by UV irradiation and transfer their excited energy to other compounds [30,31,32]. When SPL tablets are crushed or suspended, light emission efficiency to TiO_2_ or Fe_2_O_3_ might be improved, resulting in the enhancement of their photocatalytic activities. Excited TiO_2_ or Fe_2_O_3_ were able to form reactive oxygen species such as hydroxyl radicals, which might withdraw the hydrogen atom from FL followed by the formation of FL photoproduct 1. Additionally, the energy transfer from TiO_2_ or Fe_2_O_3_ to FL might promote the formation of FL photoproduct 2. The contribution of additives might be an important issue for the photochemical behavior of film-coated tablets in the case that they are crushed or suspended.

## 4. Conclusions

In conclusion, we show that the change of the dosage form of SPL tablets resulted in the decrease in its photostability, especially in the suspension. The chemical structures of two photoproducts were identified from ESI-LC/MS/MS analysis of the UV-irradiated SPL powders and suspensions. In this study, the effect of UV irradiation on the pharmacological activity and the expression of adverse effects of SPL were not evaluated. It is reported that the formation of the pyridine derivative and the pharmaceutical dimer triggered the expression of unexpected biological activities [14,33]. It is concerning that the SPL medicine, which is crushed or suspended, might have a lower pharmacological activity when it is irradiated by sunlight on account of the fact that UV is a component. The influence of photoirradiation on most medicines, when they are out of the PTP sheet and their dosage forms are changed, are not evaluated, so further research is needed that focuses on the photochemical behaviors of medicines.

## Data Availability

Not applicable.

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
