# Peer review of "Structure Determination of Felodipine Photoproducts in UV-Irradiated Medicines Using ESI-LC/MS/MS"

_pharmaceutics, 2023, doi:10.3390/pharmaceutics15020697_

Round 1

Reviewer 1 Report

Presented work is interesting and important  from photostability testing of new and existing drug point of view. 

 The main aim of the forced photodegradation studies is provide suitable information to develop and validate analytical method. Please provide full validation of the analytical method used in this studies as separate point in the results section.

Author Response

Response to Reviewer 1 Comments

Pharmaceutics-2189020

Kawabata, K., Kohashi, M., Akimoto, S., Nishi, H. Structure Determination of Felodipine Photoproducts in UV-Irradiated Medicines Using ESI-LC/MS/MS.

Thank you for your letter and the reviewer’s comments concerning our manuscript. We have studied their comments carefully and have made following revisions which we hope meet their approval.

(Point 1)

The main aim of the forced photodegradation studies is provide suitable information to develop and validate analytical method. Please provide full validation of the analytical method used in this studies as separate point in the results section.

(Response 1)

We appreciate your comment for providing the analytical insight to our manuscript. However, it is difficult to develop and validate of analytical method completely because revised manuscript must be submitted within 10 days. So, we are sorry for not accepting your comments. Before carrying out the photostability evaluation of felodipine tablets, we evaluated the type of organic solvents (methanol or acetonitrile) and the contents of them (60-90%, v/v). From obtained results, we judged that a mixture of methanol and 0.1% (v/v) formic acid (8:2, v/v) was appropriate as a mobile phase for this experiment because the analytical time was within 20 min and peak separation was well. Above sentence was added in 2.4.

We hope that revised manuscript is now acceptable for publication.

Best regards.

Your Sincerely,

NAME: Kohei Kawabata, Ph. D.

ADRESS: Faculty of Pharmacy, Yasuda Women’s University, 6-13-1 Yasuhigashi, Asaminami-ku, Hirosshima 731-0153, Japan

EMAIL: kawabata-k@yasuda-u.ac.jp

TEL: 81-82-878-9440

FAX: 81-82-878-9540

Reviewer 2 Report

This paper concerns the stability of felodipine tablets upon exposure to light. Three sample types were studied: whole commercial tablets, crushed tablets (powder sample) and crushed tablets in an aqueous medium (suspension sample).

The introduction clearly presents the background of the study and highlights the need to perform drug stability tests. The experimental section is meticulously presented. Clear figures are also an advantage of this paper (but please correct the legend in Fig. 4 B – Photoproduct 1 and 2). The results are presented in a logical order and well described.

My main concern is the aim of the study. Photostability evaluation is very important part of the formulation development and testing. However, the results show that the UV light had no effect on the tested tablets in their commercial form – UV irradiation affected only the powders and suspensions. In lines 189-192 the authors argue: “It is possible that the crush of SPL tablets for the improvement of medication compliance in a clinical situation might result in the decrease of its beneficial effect when SPL powders are packaged and stored in a place where sunlight is exposed”.

-          The experiment time (irradiation time) was 24 hours. In my opinion, if tablets have to be crushed or if a suspension has to be prepared prior to its administration, this occurs shortly before swallowing the medicine. Moreover, 24 h exposure to sunlight is practically impossible in natural conditions.

-          UV-range light constitutes only a few percent of natural sunlight. Therefore, the experiments conducted using UV-light only might not well represent the actual conditions.

Nevertheless, I find it very useful to study the photodegradation products of pharmaceuticals, as in general, trace levels of drugs are often detected in wastewater. I suggest adding at least a short paragraph about felodipine levels in aqueous environment. I think it would be interesting to refer the obtained results on the susceptibility of felodipine to photodegradation, especially in water suspension, to environmental applications of drug photodegradation studies.

I would also move some general statements from the Results & Discussion section to Conclusions.

I suggest to proofread the manuscript in search for grammar errors and typos (ex. line 73 “This is the first report [that] focused”. I also suggest using more formal expressions (ex. line 37 “So,” à “Therefore”). In my opinion, the sentence “Dihydropyridine drugs have been used for the reduction of high blood pressure but are well known as photosensitive drugs.” (lines 41-43) is not built logically (“but”: the use of dihydropyridine drugs does not have any influence on their photosensitivity). Please carefully check the whole manuscript.

Author Response

Response to Reviewer 2 Comments

Pharmaceutics-2189020

Kawabata, K., Kohashi, M., Akimoto, S., Nishi, H. Structure Determination of Felodipine Photoproducts in UV-Irradiated Medicines Using ESI-LC/MS/MS.

Thank you for your letter and the reviewer’s comments concerning our manuscript. We have studied their comments carefully and have made following revisions which we hope meet their approval.

(Point 1)

The experiment time (irradiation time) was 24 hours. In my opinion, if tablets have to be crushed or if a suspension has to be prepared prior to its administration, this occurs shortly before swallowing the medicine. Moreover, 24 h exposure to sunlight is practically impossible in natural conditions.

(Response 1)

In a clinical situation, there are several cases that tablets are crushed and packaged followed by the storage in the house of patients. Our previous reports also showed that a long-term storage (for up to 90 days) of crushed amlodipine oral disintegration tablets in the package near the window of a house, where is well exposed to sunlight irradiation, resulted in the decrease of their APIs [ref 3]. However, about most tablets, the effect of dosage form changes on their photostability is unclear. From these backgrounds, in this study, we evaluate effects of crush and suspension on the photostability of API firstly. We believe these information are important for their safe use.

(Point 2)

UV-range light constitutes only a few percent of natural sunlight. Therefore, the experiments conducted using UV-light only might not well represent the actual conditions.

(Response 2)

In this study, UV irradiation was performed using a black light. It mainly emits UV-A (most abundant wavelength is 360 nm). UV-A is the main component of UV-range light in sunlight. Furthermore, our previous study showed the photodegradation induced by UV-A irradiation was comparable to that of by sunlight irradiation [ref 16]. We think this study demonstrate the photostability evaluation in the actual conditions as much as possible.

(Point 3)

Nevertheless, I find it very useful to study the photodegradation products of pharmaceuticals, as in general, trace levels of drugs are often detected in wastewater. I suggest adding at least a short paragraph about felodipine levels in aqueous environment. I think it would be interesting to refer the obtained results on the susceptibility of felodipine to photodegradation, especially in water suspension, to environmental applications of drug photodegradation studies.

(Response 3)

We thank to your insightful comment. However, the quantification of felodipine and its photoproducts in aquatic environment is beyond the scope of our study. We would like to clarify the photochemical behavior of medicines and their altered forms. Please accept our concept.

(Point 4)

I would also move some general statements from the Results & Discussion section to Conclusions.

(Response 4)

As your suggestion, some general statements were moved from results and discussion section to conclusion section.

(Point 5)

I suggest to proofread the manuscript in search for grammar errors and typos (ex. line 73 “This is the first report [that] focused”. I also suggest using more formal expressions (ex. line 37 “So,” à “Therefore”). In my opinion, the sentence “Dihydropyridine drugs have been used for the reduction of high blood pressure but are well known as photosensitive drugs.” (lines 41-43) is not built logically (“but”: the use of dihydropyridine drugs does not have any influence on their photosensitivity). Please carefully check the whole manuscript.

(Response 5)

We checked the whole manuscript carefully.

We hope that revised manuscript is now acceptable for publication.

Best regards.

Your Sincerely,

NAME: Kohei Kawabata, Ph. D.

ADRESS: Faculty of Pharmacy, Yasuda Women’s University, 6-13-1 Yasuhigashi, Asaminami-ku, Hirosshima 731-0153, Japan

EMAIL: kawabata-k@yasuda-u.ac.jp

TEL: 81-82-878-9440

FAX: 81-82-878-9540

Reviewer 3 Report

1. Source and purity of all chemicals used should be specified in the experimental section.

2. The manuscript contains spelling/grammatical errors. So, the language should be polished thoroughly.Such as Line 88.

3. The mechanistic part should be presented as scheme in the revised manuscript.

4. Furthermore, active pharmaceutical ingredients (APIs) of several normal plain tablets, such as amlodipine, nifedipine and nilvadipine, were photodegraded when these tablets were pulverized and photo-irradiated Some related refs could be updated, such as New J. Chem., 2022, 46, 19577–19592 ; CrystEngComm, 2022, 24, 6933–6943 and Mater. Today. Commum., 2022, 31,103514 and Dalton Trans., 2021, 50, 18016–18026

5. The Reference formatting in the text is not correct. Please correct it.

6. I think the different thickness of the Splendil® (SPL) tablets is very important for the photocatalytic efficency. How could you control it?

Author Response

Response to Reviewer 3 Comments

Pharmaceutics-2189020

Kawabata, K., Kohashi, M., Akimoto, S., Nishi, H. Structure Determination of Felodipine Photoproducts in UV-Irradiated Medicines Using ESI-LC/MS/MS.

Thank you for your letter and the reviewer’s comments concerning our manuscript. We have studied their comments carefully and have made following revisions which we hope meet their approval.

(Point 1)

Source and purity of all chemicals used should be specified in the experimental section.

(Response 1)

Commercial source was described in 2.1. Purity of all chemicals were added to 2.1.

(Point 2)

The manuscript contains spelling/grammatical errors. So, the language should be polished thoroughly. Such as Line 88.

(Response 2)

We checked the whole manuscript carefully.

(Point 3)

The mechanistic part should be presented as scheme in the revised manuscript.

(Response 3)

Figure 8 was changed to Scheme 1. Furthermore, “3.3. Photodegradation Mechanism of FL” was added to results and discussion section.

(Point 4)

“Furthermore, active pharmaceutical ingredients (APIs) of several normal plain tablets, such as amlodipine, nifedipine and nilvadipine, were photodegraded when these tablets were pulverized and photo-irradiated” Some related refs could be updated, such as New J. Chem., 2022, 46, 19577–19592 ; CrystEngComm, 2022, 24, 6933–6943 and Mater. Today. Commum., 2022, 31,103514 and Dalton Trans., 2021, 50, 18016–18026

(Response 4)

Thank you for your recommendation. We check above reference and judge that they are not appropriate as reference because we would like to refer the study focused on the photodegradation of dihydropyridine tablets.

(Point 5)

The Reference formatting in the text is not correct. Please correct it.

(Response 5)

We checked the reference formatting carefully.

(Point 6)

I think the different thickness of the Splendil® (SPL) tablets is very important for the photocatalytic efficency. How could you control it?

(Response 6)

We think light emission efficiency for SPL tablet is an important issue for its photostability.

We hope that revised manuscript is now acceptable for publication.

Best regards.

Your Sincerely,

NAME: Kohei Kawabata, Ph. D.

ADRESS: Faculty of Pharmacy, Yasuda Women’s University, 6-13-1 Yasuhigashi, Asaminami-ku, Hirosshima 731-0153, Japan

EMAIL: kawabata-k@yasuda-u.ac.jp

TEL: 81-82-878-9440

FAX: 81-82-878-9540

Round 2

Reviewer 1 Report

Dear Authors

In my opinion the validation of the method is mandatory step in the forced degradation studie and it cannot be missed. So please put the short  description of validation of HPLC method.

Author Response

Response to Reviewer 1 Comments

Pharmaceutics-2189020

Kawabata, K., Kohashi, M., Akimoto, S., Nishi, H. Structure Determination of Felodipine Photoproducts in UV-Irradiated Medicines Using ESI-LC/MS/MS.

Thank you for your letter and the reviewer’s comments concerning our revised manuscript. We have studied their comments carefully and have made following revisions which we hope meet their approval.

(Point 1)

In my opinion the validation of the method is mandatory step in the forced degradation studie and it cannot be missed. So please put the short  description of validation of HPLC method

(Response 1)

Thank you for your insightful comments. We evaluated the “specificity” among validation parameters. An utilized mobile phase (0.1% formic acid : methanol = 2:8) could separate detected peaks sufficiently within 20 min as shown in Figure 2. In addition, a PDA detector was utilized for the purity check of detected peaks showing that each peak was not overlapped under the above LC condition. This sentence was added to section 2.4. We think that the evaluation of other validation parameters (accuracy, precision, detection limit, quantification limit, linearity, range and robustness) seems to be not needed in this study because our purpose is not the quantification of felodipine and its photoproducts but the detection of photoproducts. If we would like to quantify felodipine content in the tablets and altered forms accurately, the evaluation of full validation parameters must be done firstly. We judged that it is sufficient to evaluate the “specificity” for this study and described in the manuscript. Please accept our concept.

We hope that revised manuscript is now acceptable for publication.

Best regards.

Your Sincerely,

NAME: Kohei Kawabata, Ph. D.

ADRESS: Faculty of Pharmacy, Yasuda Women’s University, 6-13-1 Yasuhigashi, Asaminami-ku, Hirosshima 731-0153, Japan

EMAIL: kawabata-k@yasuda-u.ac.jp

TEL: 81-82-878-9440

FAX: 81-82-878-9540

Reviewer 3 Report

1. I think the comment on the tabllet is also important.

2. Also, the pH parameter is also an issue, pls try to work on it.

3. The current results can not support the scheme 1 : Summary of estimated FL photodegradation.

Author Response

Response to Reviewer 3 Comments

Pharmaceutics-2189020

Kawabata, K., Kohashi, M., Akimoto, S., Nishi, H. Structure Determination of Felodipine Photoproducts in UV-Irradiated Medicines Using ESI-LC/MS/MS.

Thank you for your letter and the reviewer’s comments concerning our revised manuscript. We have studied their comments carefully and have made following revisions which we hope meet their approval.

(Point 1)

I think the comment on the tabllet is also important.

(Response 1)

Several comments related to tablet samples are described in the manuscript (please refer to line 168-170, 198-200, and 215).

(Point 2)

Also, the pH parameter is also an issue, pls try to work on it.

(Response 2)

As the reviewer’s comment, we understand pH has a crucial role for the photodegradation rate and mechanism. In this study, however, we think the evaluation of pH and other factors (such as temperature and reactive oxygen species) are not investigated because our purpose is not the elucidation of kinetic constants. We would like to give the information for readers that felodipine tablets are photodegradable when they are crushed or suspended, those are often done in a clinical situation.

(Point 3)

The current results can not support the scheme 1 : Summary of estimated FL photodegradation.

(Response 3)

In this manuscript, we determined chemical structures of two felodipine photoproducts by ESI-LC/MS/MS as shown in Figure 6-7. Based on their structures, we speculated that photoproduct 1 is generated by the oxidation of a dihydropyridine ring and photoproduct 2 is generated by the dimerization of two felodipine molecules. Scheme 1 summarizes that felodipine is converted to elucidated two photoproducts. We think Scheme 1 makes it easy for readers to understand the photochemical behavior of felodipine. We hope the reviewer accepts our concept.

We hope that revised manuscript is now acceptable for publication.

Best regards.

Your Sincerely,

NAME: Kohei Kawabata, Ph. D.

ADRESS: Faculty of Pharmacy, Yasuda Women’s University, 6-13-1 Yasuhigashi, Asaminami-ku, Hirosshima 731-0153, Japan

EMAIL: kawabata-k@yasuda-u.ac.jp

TEL: 81-82-878-9440

FAX: 81-82-878-9540

Round 3

Reviewer 3 Report

The author still not replied my comments, I can not try to review this paper.

Author Response

Response to Reviewer 3 Comments

Pharmaceutics-2189020

Kawabata, K., Kohashi, M., Akimoto, S., Nishi, H. Structure Determination of Felodipine Photoproducts in UV-Irradiated Medicines Using ESI-LC/MS/MS.

Thank you for your letter and the reviewer’s comments concerning our revised manuscript. We have studied their comments carefully and have made following revisions which we hope meet their approval.

(Point 1)

The author still not replied my comments, I can not try to review this paper.

(Response 1)

As mentioned in 2nd revised manuscript, several comments related to tablet samples are already described in the manuscript (please refer to line 168-170, 198-200, and 215). Secondly, we described the statement about the effect of pH value on felodipine photodegradation in results and discussion section (please refer to line 304-307). In this study, however, we think the evaluation and the effect of pH are not investigated because our purpose is not the elucidation of kinetic constants. Finally, a following sentence was added in results and discussion section, “We thought two FL photodegradation pathways were independent because two FL photoproducts were not generated step by step as shown in Figure 2.”, to response to reviewer response 3 (please refer to line 282-284). In addition, we previously described that FL photoproduct 1 is generated by the oxidation of a dihydropyridine ring and FL photoproduct 2 is generated by the dimerization of two felodipine molecules. Scheme 1 summarizes that felodipine is converted to elucidated two photoproducts independently. We think Scheme 1 makes it easy for readers to understand the photochemical behavior of felodipine. We hope the reviewer accepts our concept strongly.

We hope that revised manuscript is now acceptable for publication.

Best regards.

Your Sincerely,

NAME: Kohei Kawabata, Ph. D.

ADRESS: Faculty of Pharmacy, Yasuda Women’s University, 6-13-1 Yasuhigashi, Asaminami-ku, Hirosshima 731-0153, Japan

EMAIL: kawabata-k@yasuda-u.ac.jp

TEL: 81-82-878-9440

FAX: 81-82-878-9540

Round 4

Reviewer 3 Report

I have not much report.

Author Response

Pharmaceutics-2189020

Kawabata, K., Kohashi, M., Akimoto, S., Nishi, H. Structure Determination of Felodipine Photoproducts in UV-Irradiated Medicines Using ESI-LC/MS/MS.

Thank you for your letter and the reviewer’s comments concerning our revised manuscript. We have studied their comments carefully and have made following revisions which we hope meet their approval.

(Point 1)

I have not much report.

(Response 1)

We deeply appreciate to your insightful comments.

We hope that revised manuscript is now acceptable for publication.

Best regards.

Your Sincerely,

NAME: Kohei Kawabata, Ph. D.

ADRESS: Faculty of Pharmacy, Yasuda Women’s University, 6-13-1 Yasuhigashi, Asaminami-ku, Hirosshima 731-0153, Japan

EMAIL: kawabata-k@yasuda-u.ac.jp

TEL: 81-82-878-9440

FAX: 81-82-878-9540
